# Living Arrangement Intentions of Adult Migrant Children toward Their Left-Behind Rural Parents in China

**Jianxi Feng** [1],* and **Shuangshuang Tang** [2],*

[1]  School of Architecture and Urban Planning, Nanjing University, Nanjing 210093, China
[2]  Jiangsu Center for Collaborative Innovation in Geographical Information Resource Development and Application, School of Geography, Nanjing Normal University, Nanjing 210023, China
*   Correspondence: jxfup@nju.edu.cn (J.F.); sstang@njnu.edu.cn (S.T.)

**Abstract:** The number of "left-behind" rural elderly has been increasing in China. Although rural elderly support has been examined, the existing studies mainly reveal the current living arrangements of the rural elderly and the actual support that adult migrant children provide for their old parents. The rural migrants' intended living arrangements for their "left-behind" parents have rarely been investigated. Therefore, the paper aims to investigate the pattern of and explore the underlying decision-making involved in living arrangements that adult migrant children intend to provide for their "left-behind" aging parents in the countryside. It is assumed that living arrangements were the functions of adult migrants' residency plan for the future, their socioeconomic status and cultural values towards "filial piety". A questionnaire survey was conducted in Jiangsu province to test the assumptions. One-way ANOVA and Tukey post-hoc tests were used to examine the disparities of the socioeconomic characteristics among the seven options of living arrangements. The results show that most of the rural migrants wanted to change their parents' current "left-behind" status. They preferred to live together with their elderly parents in rural hometowns. A significant proportion of the respondents consider settling in cities adjacent to their rural hometowns. Rural migrants' intended living arrangements for their left-behind parents are mediated by their future expectations, caring attitudes and financial circumstances. Institutional barriers for, and the disadvantageous status of, the rural migrants are the main impact factors.

**Keywords:** living arrangement intention; old-age support; migrant children; rural elderly; China

## 1. Introduction

China has undergone dramatic development since the economic reforms started in the late 1970s. During the transitional period, urbanization and industrialization have driven many young rural workers to seek employment in cities, leaving their parents behind in the countryside [1–3]. The term "left-behind" is used to describe the family members of those rural migrants who remain in the place of origin [4,5]. In 2015, there were 277 million rural migrants in urban areas [6] and around 50 million left-behind elderly in rural areas [5]. Subject to the increasing life expectancy and the persistent outflow of the young rural workforce, the number of "left-behind" rural elderly will increase drastically [7,8].

Unlike Western industrialized countries, where pension and healthcare systems for the elderly are well-developed, public social security funds are often insufficient to meet the needs of the rural elderly in the developing world [9]. Family members, particularly adult children, are used to playing a vital role in supporting the rural elderly [10]. However, massive rural-to-urban migration has distorted the traditional co-residence pattern of parents and adult children, impeding caregiving for the rural elderly [11]. Given the growing number of "left-behind" rural elderly and the persistent underdeveloped healthcare and social insurance systems in rural areas, it is imperative to investigate the adult migrant children's intended living arrangement for their "left-behind" aging parents and also their challenges in providing old-age support.

In the scholarly realm, researchers are increasingly concerned about living arrangements and caregiving for the rural elderly [9,12,13]. Several studies have examined the topic from the perspective of the left-behind elderly in developing countries [1,14], particularly in China [15,16], focusing on how the provision for the left-behind elderly is changed once their children have moved to urban areas. Nevertheless, little research has explicitly investigated the topic from the perspective of adult migrant children in developing countries, especially regarding what they are going to provide to their left-behind aging parents in the future. Existing research has demonstrated that the living arrangement and caregiving of aging parents are profoundly influenced by the adult children's future plan [17,18]. Given that China is still experiencing dramatic changes in both the living status of the rural–urban migrants and the rural public social services regarding elderly caregiving, the living arrangements and provision of old-age support to the left-behind parents will change accordingly. It is therefore necessary to investigate how adult migrant children plan to care for their left-behind parents.

Apart from individual-level and household-level factors, the living arrangement and caregiving for the rural elderly are mediated by China's rapidly changing social environment. For instance, Chinese culture is rooted in Confucianism, which is grounded on a rigid ethical system to regulate individual behaviours and interpersonal relationships [19]. The Confucius teaching of xiao (filial piety, showing respect to one's parents, elders and ancestors) urges the young to meet the material and emotional needs of their parents. The rapid socioeconomic development in China has challenged traditional norms in many ways [19], which may significantly affect the provision received by the rural elderly. A significant portion of migrant children might change the current leaving-behind status of their aging parents and select another mode to support them in the future [20]. Thus, exploring adult migrant children's plans for their left-behind parents may provide a more holistic view of the support for the rural elderly in China. The associated findings are imperative for understanding and predicting the living arrangements of the rural elderly in the future. In addition, China is entering a fast- aging society in which the rural elderly account for a large proportion. The research on the intended living arrangement of adult migrant children toward their left-behind rural parents may provide hints about where those adult migrant children and their left-behind rural parents would ultimately go, which will reshape the urban and rural population's structures and relationships.

To fill the above research gaps, this study seeks to (1) investigate the pattern of living arrangements that adult migrant children intend to provide for their "left-behind" aging parents in the countryside, and (2) explore the underlying decision-making involved in living arrangements associated with elderly care by comparing the disparities among different subgroups. A recent questionnaire survey of rural migrants was conducted in Jiangsu province in 2015 to address the above issues. One-way ANOVA and Tukey post-hoc tests were used to examine the disparities of the socioeconomic characteristics among the seven options of living arrangements. This paper is structured as follows: Section Two presents a review of living arrangements and elderly support; Section Three describes the study area and the survey; Section Four compares and analyses the characteristics of different subgroups; and Section Five summarizes the conclusions.

## 2. Living Arrangements and Elderly Support in China

### 2.1. Living Arrangements from the Perspective of the Elderly

Living arrangements are essential in the later stages of the lifecycle and are closely related to old-age support [17]. Existing studies have explored the patterns and determinants of the living arrangement of the elderly. Various living patterns, such as solitary living at home, co-residence with children and being institutionalized, are often mentioned [9,20]. Concerning the determinants, the socioeconomic attributes of the elderly, such as age, gender, occupation, and health status, significantly influence their living arrangements [3]. Moreover, it is observed that social norms and personal preferences of the elderly also affect their living arrangements [17].

Unlike many advanced western countries where independent living among the elderly is ordinary, co-residence with children is a viable and pervasive pattern in Asian countries [12,21]. Nevertheless, more living arrangement options appear with changes in individual features and socioeconomic settings [21]. In Thailand, with increased income, the elderly may become more self-supporting and pursue greater privacy in living arrangements [13]. Moreover, the transitions in sociocultural norms might play a role in the living arrangement of the aged [12]. It has been stated that modernization following industrialization and urbanization might change elderly living arrangements based on the traditional family system [9]. On the one hand, the modernization process undermines extended family relations and their function for old-age care [22]. On the other hand, the young labour force might become unwilling to support their older parents when they move into a modern society.

In China, family support and rural land have guaranteed the subsistence of the rural elderly for centuries [23]. After the establishment of the People's Republic of China, the government established a dual-track public social welfare system, whereby the work unit in urban areas provided the social benefits of urban workers, and the responsibility for social welfare in rural areas was primarily left to the village collectives (communes) [24]. Since 1978, economic reforms have broken down the former social security systems. Specifically, in the countryside, the household responsibility system affords individual households the freedom to manage their allocated plots of land and thus increase their income. Nevertheless, the collapse of communes, which were based on collective management, has returned the responsibility for rural old-age support to rural populations. To tackle the growing problem of rural old-age support, the state introduced a rural pension scheme in the 1990s; later, in 2009, it introduced a new rural social pension scheme. Unlike the former scheme, which was funded by individual payments, the more recent scheme is funded by individual payments and subsidies from rural collectives and governments [23]. Nevertheless, the current rural pension is low, and most of the rural elderly are still in financial difficulty and living in poor conditions [7]. Therefore, family members, particularly adult children, play a primary role in elderly care in China.

### 2.2. The Elderly Support Provided by Adult Migrant Children

Migration often breaks the former living patterns between children and parents and constructs a new mode of old-age support. Among the related literature, there is minimal research concerning the relationship between adult children's outflows and care for older rural parents in the developing world [16]. Several studies have focused on the support patterns provided to the elderly by migrant children in developing countries. Generally, elderly support commonly comprises several types, such as instrumental support, financial support, and emotional support. During the out-migration period, the decline in the co-residence of rural parents and adult children reduces the instrumental support for aging parents [23]. However, adult migrant children can care for their parents by providing financial and emotional support in the form of remittances and contact.

Regarding the determinants, it is observed that living proximity, the characteristics of older adults and the life circumstances of adult children affect old-age support [25]. Typically, people make the most of the possibilities available to them [9]. For instance, poor human capital and low living costs increase the co-residency of elderly parents and their children [5]. Moreover, the socioeconomic characteristics of migrant children, such as age, educational attainment and income, significantly impact on old-age support [11,16,25]. For instance, migrant children who receive decent jobs at their settlement destinations provide material benefits for their parents. In contrast, migration of low-paid children has little economic impact on elder caregiving [14]. In addition, elderly support is gender-oriented in some societies. Some researchers have demonstrated that the support provided to parents by migrant children comes mainly from sons rather than daughters [16,26].

The One-Child policy has been reported to have had considerable influences on the living arrangements and elder care in both urban and rural areas in China in multiple

ways [27,28], including the acceleration of population aging, the changes to family and kinship structure and the norms of family and intergenerational relationships. Large extended families, where elders are supported and youngsters are cared for under the same roof, have never been the major family pattern in rural China [28]. However, with the adoption of the One-Child policy, each family in rural areas can only have one or two children (according to the policy, if the first child born in rural areas is a girl, then farmers may wait four years and have a second child), household size has decreased dramatically and the adult migrants' burden of elder care has increased [29]. In rural areas, traditional values associated with filial piety remain strong, and gendered parent–child relations are observed [27]. In the Chinese patrilineal family system, sons are primarily responsible for staying with their parents and caring for them in old age as they continue the family line and contribute to ancestral rituals [26]. Daughters are the second choice [29].

In China, the socioeconomic contexts have gradually changed the behaviours and attitudes of adult migrant children toward old-age support. On the one hand, decades of massive rural-to-urban migration have greatly impacted on living patterns and old-age support in the countryside. The exodus of the young from rural areas has resulted in their geographical separation from their elderly parents, impeding close intergenerational exchanges [30]. Moreover, rural migrants commonly suffer from poor conditions. They have difficulties in settling down in cities [31], which reduces the ratio of co-residence in destination cities and the level of support they can provide to their parents. On the other hand, the modernization process brought about by the economic reforms of 1978 and the more recent urbanization seems to have changed the attitudes of the young toward the old. In the transitional period, the family authority has shifted from the old heads of the family to the young ones in some parts of rural areas in China, and the attitudes of the young toward the old are changing as society becomes increasingly individualistic and market-oriented [4].

In sum, although several studies have examined living arrangements from the perspective of the present support that adult migrant children provide for their old parents, the living arrangement intentions of adult migrant children toward their "left-behind" rural elder parents have rarely been examined. It has to be noted that adult migrant children might change their current "leaving-behind" status when their rural parents are old. In addition to exploring the current living pattern, it is necessary to examine the changes in living arrangements in the future because this may reshape the structure of elderly care provision and the distribution pattern of the floating population in China. Furthermore, such intentions of living arrangements might be influenced not only by the features of adult migrant children but also by the rapidly changing socioeconomic context, which might deepen the understanding of the mechanism of living arrangements and the associated elder caregiving. Given the research and practical significance of this issue, the living arrangement intentions of adult migrant children toward their "left-behind" elderly parents require further study.

## 3. Research Design and Conceptual Framework

This paper aims to investigate the pattern of and explore the underlying decision-making involved in living arrangements that adult migrant children intend to provide for their "left-behind" aging parents in the countryside. It is assumed that living arrangements were the functions of adult migrants' residency plan for the future, their socioeconomic status and cultural values towards "filial piety". The following data, methods and conceptual framework are used to test the assumption.

### 3.1. Study Area

The 2015 questionnaire survey on which the present research is grounded was conducted in two megacities (Nanjing and Suzhou) in Jiangsu province. The two cities were selected for two reasons. First, the two cities are major economic centres in one of China's most developed regions. Consequently, they are prominent destinations for rural migrants

from different parts of China, including inland provinces. Nanjing, the provincial capital of Jiangsu province, administrates an area of 6587 km² [32]. The GDP per capita of Nanjing ranked the third in Jiangsu. In 2015, 1.73 million of the 8.24 million residents were migrants [32]. Due to its proximity to Shanghai, Suzhou has developed a solid export-oriented economy. In 2015, the GDP per capita of Suzhou ranked first in Jiangsu Province. Suzhou has 10.62 million residents and an area of 8657 km² [32]. Because of the rapid economic development, abundant employment opportunities have attracted a large population of migrants. Approximately 40 percent of the population in Suzhou were migrants [32].

Second, the two cities have different economic structures. With the economic restructuring in recent decades, the tertiary industry in Nanjing already accounted for 57% of the economy in 2015, while the secondary industry still accounts for 49% of the gross domestic product in Suzhou [32]. The disparities in the industrial structures of the two cities determine some distinct characteristics of the two cities and their labour forces. In sum, the presence of a large number of rural migrants in the two cities and the structural disparities of their labour market provided a large and diverse sample pool for exploring the living arrangement intentions of rural migrant children toward their "left-behind" parents in the countryside.

### 3.2. Data and Methods

The empirical analysis uses the data from a 2015 questionnaire survey of rural migrants in Nanjing and Suzhou of Jiangsu. From May to November 2015, the survey was conducted by the Jiangsu Provincial Bureau of Statistics via face-to-face interviews. A stratified sampling method was adopted for the survey. First, five inner-city districts and two suburban districts were chosen from the 11 districts in Nanjing. Three inner-city districts, one suburban district, and one county-level city were selected from the five districts and four county-level cities in Suzhou. Second, the questionnaires were distributed in the chosen districts and county-level cities. The Sixth National Census was employed to construct the sampling frame, focusing on the major sectors that hire rural migrants. According to the census, rural migrants were mainly engaged in the manufacturing, construction and service sectors. In the survey, two sampling methods were used, one for the manufacturing and construction sector and one for the service sector. Due to the geographic clustering of the manufacturing and construction activities, factories and construction sites are mainly found in certain parts of Nanjing and Suzhou. Thus, based on the list of factories and construction sites in the chosen districts and county-level cities in Nanjing and Suzhou and the Sixth National Census, a sampling pool consisting of the 50 industrial areas and 30 construction sites with the highest concentration of rural migrants was formed. From the sampling pool, 18 industrial areas and eight construction sites were randomly selected. As for the service sector, 22 urban neighbourhoods from all of the chosen districts and county-level cities were selected based on the density of rural migrants and the distribution of economic workplaces that accommodated rural migrants in that district or county-level city. At each sampled site, in reference to the proportion of rural migrants in that area provided by the census, between 20 and 25 respondents were randomly selected to participate in the survey. In total, 1065 questionnaires were distributed, with 624 samples in Suzhou and 441 samples in Nanjing. The response rate of the survey was 92%. The sample size accounts for approximately 0.02% of the total migrant population in each of the two cities. The representativeness of the surveyed samples is largely consistent with the migrant population.

As the aim of this study was to examine the living arrangement intentions and elderly support of adult migrant children whose parents were left behind in the countryside, respondents younger than 18 years old, and respondents whose parents were either dead or lived in places other than their rural hometowns were excluded. This left 873 valid cases for analysis.

To examine the significance of differences in the socioeconomic characteristics among the different living arrangements options, one-way ANOVA tests and specifically, Tukey post-hoc tests were adopted in the analyses.

### 3.3. Conceptual Framework

As mentioned above, living arrangement reflects intergenerational relations, which significantly affects old-age support [17]. Previous studies have demonstrated that adult migrant children's intended future living arrangement for their left-behind parents is influenced by their residency plan for the future and cultural values towards "filial piety" (i.e., the perceived responsibility for providing elderly support) [5,11,14]. In addition, the socioeconomic status of both the adult children and the elderly not only exert a directly influence on the intended living arrangement of the left-behind parents, but also indirectly influence the living arrangement through the path of settlement choices and attitudes toward elderly care of the adult children. Particularly, as the parent–child relations are highly gendered in rural areas, the influences of sex are included and discussed. The conceptual framework of these linkages is presented in Figure 1, and the data analyses were conducted to follow it. Given that the information of the socioeconomic status of the elderly was not collected, this paper only examines the relationships between the adult migrant children's socioeconomic status, their future residency plan, perceived responsibility for providing elderly support, and intended living arrangement for the left-behind elder parents.

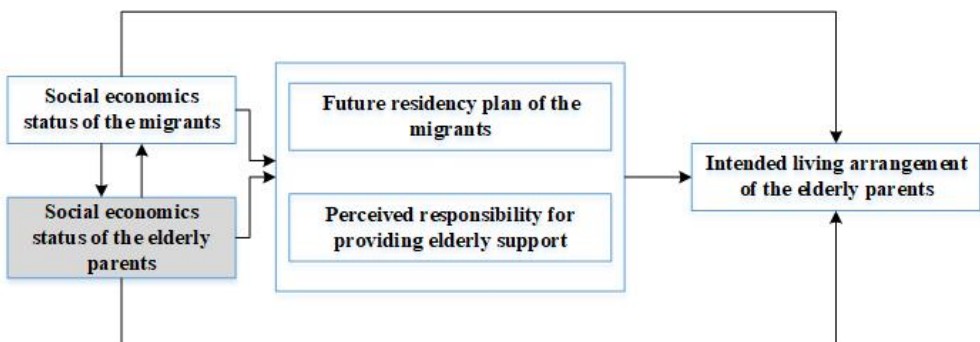

**Figure 1.** Conceptual framework of the migrants' intended living arrangement for their elder parents.

## 4. Results and Discussion

### 4.1. Intended Living Arrangements for the Left-Behind Parents

In this study, the respondents were classified into several groups based on their responses to a multiple-choice question in the questionnaire, namely: "Which living arrangement do you intend to adopt for your 'left-behind' parents when they are old?" There are seven options (see Table 1). In the table, "urban area" indicates the host cities where the adult migrants work and live and "rural hometown" refers to the county seat or township of their original village. In general, the vast majority of respondents (94.3%) intended to change the current left-behind status of their parents. Among those respondents, most (68.2%) selected the traditional "co-residence" pattern to support their left-behind parents, whereas the proportions of those choosing other options remained low. Specifically, 39.9 percent intended to co-reside with their elderly parents in an urban area, and 28.3 percent planned to live together in their rural hometown. Slightly more than 10 percent of respondents wanted to live separately in the same urban destination (7.6%) or village of origin (5.3%); 5.7 percent planned to maintain the status quo in the future; 9.5 percent reported that their elderly parents would live with their siblings or other relatives. Finally, very few (3.7%) intended to put their elderly parents in nursing homes.

**Table 1.** Adult migrant children's intended living arrangements for their left-behind parents.

| Intended Living Arrangements for Left-Behind Parents | Number | Percentage |
|---|---|---|
| Living together in urban area | 348 | 39.9 |
| Living separately in the same urban area | 67 | 7.6 |
| Living together in rural hometown | 247 | 28.3 |
| Living separately in the same village | 46 | 5.3 |
| Continue to leave parents behind in rural hometown (status quo) | 50 | 5.7 |
| Sending them to nursing homes | 32 | 3.7 |
| Sending them to other siblings or relatives | 83 | 9.5 |
| Total | 873 | 100.0 |

*4.2. Influences of Socioeconomic Characteristics on the Intended Living Arrangements*

According to the conceptual model, the disparity of the intended living arrangement among the respondents may be associated with their socioeconomic attributes. To explore the underlying decision-making regarding living arrangements and associated old-age care among the seven subgroups shown in Table 1, several attributes of adult migrant children were considered in this study.

Socioeconomic factors such as human capital (age, education attainment), social attributes (gender, marital status) and economic status (income, occupation) are associated with migrants' old-age care for their parents [11,20]. Based on those studies, several socioeconomic factors were chosen here (see Table 2). One-way ANOVA tests were used to examine the disparities of the socioeconomic characteristics among the seven subgroups. The results are shown in Table 2. A $p$-value of less than 0.05 indicates a statistically significant relationship. After running the one-way ANOVA tests, Tukey post-hoc tests were then adopted to compare different subgroups.

**Table 2.** Socioeconomic characteristics of the respondents of the seven subgroups (percent).

| | Together in Urban Areas | Separately in Urban Areas | Together in Rural Areas | Separately in Rural Areas | Status Quo | Nursing Home | Care by Siblings or Relatives | Total | F-Value | *p*-Value |
|---|---|---|---|---|---|---|---|---|---|---|
| **Human Capital** | | | | | | | | | | |
| Age (mean) | 31.1 | 28.6 | 33.2 | 34.8 | 37.4 | 36.4 | 35.1 | 32.6 | 8.66 | 0.000 |
| **Educational attainment** | | | | | | | | | | |
| Primary or below | 8.3 | 1.5 | 13.6 | 35.0 | 10.0 | 9.4 | 10.8 | 10.4 | 3.94 | 0.001 |
| Middle school | 50.6 | 16.4 | 69.6 | 50.0 | 62.0 | 59.4 | 63.9 | 56.2 | 12.69 | 0.000 |
| College and above | 41.1 | 82.1 | 16.8 | 15.0 | 28.0 | 31.2 | 25.3 | 33.4 | 23.09 | 0.000 |
| **Social attribute** | | | | | | | | | | |
| Male | 52.3 | 62.7 | 67.4 | 70.0 | 50.0 | 59.4 | 42.2 | 57.4 | 4.43 | 0.000 |
| Married | 49.7 | 29.9 | 66.3 | 60.0 | 74.0 | 65.6 | 69.9 | 57.5 | 8.76 | 0.000 |
| **Economic status Income (RMB)** | | | | | | | | | | |
| <4000 | 42.5 | 37.3 | 53.5 | 30.0 | 46.0 | 68.8 | 56.6 | 46.0 | 2.12 | 0.049 |
| 4000–6000 | 46.3 | 52.2 | 41.0 | 65.0 | 46.0 | 21.9 | 30.1 | 44.7 | 1.53 | 0.166 |
| >6000 | 11.2 | 10.5 | 5.5 | 5.0 | 8.0 | 9.3 | 13.3 | 9.3 | 1.52 | 0.168 |
| **Job type** | | | | | | | | | | |
| Manual worker | 57.2 | 40.3 | 63.4 | 60.0 | 42.0 | 43.8 | 47.0 | 55.6 | 3.66 | 0.001 |
| Sales and services | 29.3 | 23.9 | 33.0 | 30.0 | 42.0 | 37.4 | 37.3 | 31.8 | 1.20 | 0.305 |
| Office worker | 13.5 | 35.8 | 3.6 | 10.0 | 16.0 | 18.8 | 15.7 | 12.6 | 9.76 | 0.000 |

Note: dependent variable = intended living arrangement for the left-behind rural parents.

Almost all of the socioeconomic factors are statistically different among the seven subgroups. Concerning age, results of the Tukey post-hoc test showed that respondents

who intended to support their elderly parents in an urban area are significantly younger than those who planned to care for their elderly parents in their rural hometown, maintain the status quo or put their parents in a nursing home. Typically, young people prefer to settle in urban areas [31], which impacts their living arrangements for their parents. Moreover, compared with older migrants, younger migrants generally have more human capital and expectation for bringing their parents to cities.

After running a Tukey post-hoc test, it was found that respondents who chose urban areas as support places were significantly higher-educated than those who intended to return to the village to provide care for their elderly parents. Among the respondents, those who intended to have their parents live in different places in the same city had the highest educational attainments. High human capital could help afford the cost of urban housing and provide high financial capacity for adopting separate living patterns within the same city. Moreover, highly educated migrants might be influenced more by modern culture and, thus, may choose a modern option whereby elderly parents and adult children do not live together but live near one another to keep intergenerational linkages.

In terms of social attributes, female respondents tended to arrange their parents to live with their siblings or other relatives. In contrast, a larger proportion of the male respondents chose the traditional co-residence living arrangement. Siblings play an important role in parents' support. China's family planning policies have followed a dual track along the entrenched rural–urban divide [26]: if the first child born in rural areas is a girl, farmers may wait four years and have a second child. This means that most of the migrants have siblings. According to Chinese tradition, sons are more responsible for caring for their parents than daughters [27]. However, due to the enhancement of the power of women in the labour market and the changes in social and cultural norms, women tend to play more roles in financial and emotional support to elder care, though sons are still the primary providers and tend to live tother with their aged parents. After running a Tukey post-hoc test, it was further found that female respondents intended to co-reside with their elder parents in urban areas rather than in the countryside. Rural migrants who intended to return to their home village and keep their rural roots were more persist to traditional norms. Another explanation is that co-residence with the daughter is not pervasive for older people in the countryside. The traditional norms remain pervasive among rural migrants, even though many of them are young and influenced by modern culture.

Regarding marital status, respondents who are single preferred to live separately from but in the same city as their parents. It is possible that living in proximity to the same city is an ideal and modern way for rural migrants, especially young rural migrants, to care for their parents. Nevertheless, if they are married, they may be older and, thus, might have stronger traditional attitudes toward elderly care or have more obstacles that may affect their support plans.

Concerning economic status, respondents who intended to care for their parents in an urban area were significantly better paid than those who planned to support their parents in the countryside. This is probably because income is positively related to urban residency [33], which improves the likelihood of living in cities.

As expected, a larger proportion of manual workers intended to support their parents in their village of origin (62.1%). In contrast, more office workers planned to live separately from their elderly parents (35.8%) or put them in nursing homes (18.8%). Respondents who intended to live separately in urban areas are significantly more likely to be white-collar workers than the other subgroups. It may be that, compared to manual workers, office workers have the more abundant human capital to settle down in cities and hence opt for modern customs of elderly care. According to the interviews, respondents with higher human capital tended to behave more like their urban peers. For instance, when asked about the reasons for choosing to live separately rather than co-residence, the most common answers included "avoiding family conflicts" and "enjoying personal life". In addition, sending parents to nursing homes is generally considered as disobedient behaviour in traditional Chinese society, as it is opposite to the traditional notion that the family functions

as a stable unit of several generations under one roof. Thus, rural migrants commonly follow traditional customs to avoid public criticism. Nevertheless, the white-collar rural migrant is prone to demonstrate how strongly their conception is influenced by modern culture and tends to be more open to the nursing home.

*4.3. Influences of Intended Residence on the Intended Living Arrangements for Their Parents*

As mentioned, the future residency plan of rural migrants is an essential factor influencing their living arrangements and care plan to aged parents. In this study, the analysis of migrant children's intended residence was based on a question in the questionnaire: "Where do you plan to live in the future?" The intended residence of rural migrant children is shown in Table 3. An ANOVA test was employed to examine the disparities in the intended residence among the seven subgroups. The *p*-value (less than 0.001) indicates that the intended residence significantly differs among the seven subgroups.

**Table 3.** Adult migrant children's intended residence (percent).

| | Together in Urban Areas | Separately in Urban Areas | Together in Rural Areas | Separately in Rural Areas | Status Quo | Nursing Home | Care by Siblings or Relative | Total | F-Value | *p*-Value |
|---|---|---|---|---|---|---|---|---|---|---|
| Rural hometown | 32.5 | 13.4 | 76.2 | 85.0 | 34.0 | 32.4 | 43.4 | 47.1 | 35.06 | 0.000 |
| City or town near rural hometown | 22.7 | 41.9 | 16.0 | 10.0 | 32.0 | 38.3 | 31.3 | 23.8 | 5.60 | 0.000 |
| Current city or town | 42.8 | 40.4 | 6.5 | 5.0 | 34.0 | 29.4 | 24.1 | 27.5 | 18.76 | 0.000 |
| Other places | 2.0 | 4.5 | 1.3 | 0.0 | 0.0 | 2.9 | 1.2 | 1.6 | 1.61 | 0.141 |

Note: dependent variable = intended living arrangement for the left-behind rural parents.

As expected, the overwhelming majority of respondents who want to arrange for their parents to live in rural areas intended to settle in their rural hometowns. In contrast, for those who wanted to co-reside with their parents in urban areas, only 32.5% chose rural hometowns, while 22.7% chose the cities near their rural hometowns. For them, co-residing with parents in an urban area is out of practical concern. During the interviews, many respondents mentioned that they still needed to work in cities for a higher income. As the living cost in big cities is relatively high, co-residence with parents in small cities is an economically viable option. Moreover, the parents can do some housework at home to support their migrant children, such as preparing meals and taking care of grandchildren. This can further compensate for the high living cost in urban areas.

Of the respondents who chose "living separately with parents in urban areas," 41.9% selected cities or towns near their rural hometowns as their intended residence. For this group, cities or towns near their hometown are ideal places for living separately and yet, close to their parents. In addition, the living costs and institutional hurdles in those cities and towns are lower when compared with big cities, which may help the respondents to achieve their desired living pattern.

Most of the respondents who planned to send their parents to other siblings or relatives said they intended to settle down in their rural hometowns (43.4%) or cities near their rural hometowns (31.3%). This is common for females in rural China [26]. In traditional custom, sons have more responsibilities for supporting their parents than daughters do. However, daughters still have to take care of their parents regularly as an expression of xiao. Thus, rural hometowns or cities near rural hometowns are ideal places for women to fulfil their xiao [34]. It could be viewed as a living arrangement compatible with traditional values. For instance, some female respondents stated that it was convenient to support their parents if they lived nearby. Even though their parents co-resided with their brothers, they would go there to take care of their parents regularly.

Compared to the group planning to look after their parents in rural areas, a significantly larger share of the respondents who intended to maintain the status quo wanted to settle

in urban areas (66.0%). For this group, their pursuit of urban residence might maximize their benefits but somehow reduce their intentions to change their current living pattern to support their parents.

### 4.4. Influences of Perceived Responsibility for Providing Elderly Support on the Intended Living Arrangements

According to the conceptual model, the future living arrangements of the aged parents are also influenced by the migrants' attitudes towards the responsibility for providing elderly support. In this subsection, the seven subgroups' current support behaviours were firstly compared to reveal their intended living arrangements and elderly care in the future. Here, two types of current support behaviour by adult migrant children (financial and emotional support) are illustrated based on two questions in the questionnaire: "how much money (RMB) do you give your parents every year?" and "how often do you make contact with your parents in the rural hometown?". The possible responses to the latter question were (1) less than twice a month, (2) two or three times a month, (3) once a week, (4) two or three times a week and (5) every day.

One-way ANOVA tests and Turey post-hoc tests were performed to compare the current financial and emotional support to parents provided by the seven subgroups. The tests were based on the responses to the two questions mentioned above, which were designed to measure the degrees of current support behaviours. Concerning financial support, higher scores indicate that the migrant children gave more money (RMB) to their left-behind parents. For the emotional form of intergenerational relations, higher scores indicate higher contact frequency between migrant children and their left-behind parents. The results are shown in Table 4.

**Table 4.** Adult migrant children's current financial and emotional support to their left-behind parents.

| Category | Financial Support | | Emotional Support | |
|---|---|---|---|---|
| | Mean (RMB) | SD | Mean | SD |
| Together in urban areas | 5050.57 | 527.50 | 3.47 | 1.14 |
| Separately in urban areas | 9641.12 | 1272.54 | 3.52 | 1.07 |
| Together in rural areas | 2663.37 | 375.45 | 3.18 | 1.08 |
| Separately in rural areas | 5600.00 | 865.72 | 3.25 | 1.31 |
| Status quo | 2564.00 | 865.63 | 2.74 | 1.38 |
| Nursing home | 4567.64 | 1437.35 | 3.48 | 1.37 |
| Care by siblings or relatives | 4650.60 | 866.73 | 3.18 | 1.17 |
| Total | 4490.44 | 288.18 | 3.30 | 1.22 |
| F-value | 9.35 | | 4.26 | |
| p-value | 0.000 | | 0.000 | |

Note: dependent variable = intended living arrangement for the left-behind rural parents.

One-way ANOVA tests indicate significant differences in financial and emotional support among the seven subgroups. After running a Tukey post-hoc test, it was further found that respondents who intended either to maintain the status quo or to co-reside with elderly parents in their rural hometown provided a significantly lower level of remittances for their parents than those who planned to support elder parents in an urban area. Those who planned to live separately from but in the same city as their parents gave their parents significantly more money than the other sub-groups. The possible reason for this is that rural migrants make rational choices based on their financial capacity. For instance, the well-paid group can offer their parents more financial support and pursue a more ideal living pattern. In contrast, the low-paid group must either maintain the status quo, co-reside with parents or return to their village of origin to reduce living costs.

For emotional support, the results of Tukey post-hoc test showed that respondents who intended to care for their parents in urban areas contacted their parents more frequently than those who chose rural living patterns. Among all respondents, those who planned to maintain the status quo had the lowest contact frequency. As described above, those

aiming to care for their rural parents in cities tended to be those with more social capital and financial status. They were more willing to use new communication technologies to maintain their emotional ties with their parents. According to the interviews, apart from making telephone contact, the young, educated migrants tended to use new communication tools such as Wei Xin and QQ (the main internet chat tools in China). The improvement in communications somewhat enables family members to maintain emotional ties and social exchanges, even if they do not live near each other [35]. On the other hand, those planning to support their parents in the countryside were relatively less educated and in lower social ranks. This social group is less likely to make use of new communication technologies through the Internet, while telephone contact is often expensive for them, which somewhat limits contact frequency. The behaviours of those intending to maintain the status quo might be partly explained by their marginal positions in cities and an increasing individualism when they move into a modern society.

After examining the current behaviours of old-age care, the adult migrant children's perceived responsibility for providing elderly care was analysed. Here, the attitudes toward elderly care are illustrated based on a question in the questionnaire, namely, "who do you think is responsible for old-age care?". The question comes with three answers (1: elderly themselves; 2: children; 3: government), each with three degrees: (1: no responsibility; 2: partial responsibility; 3: full responsibility). One-way ANOVA and Tukey post-hoc tests were used to reveal the degrees selected by the different support subgroups. Higher scores indicate more responsibility for elderly support. The results are shown in Table 5.

**Table 5.** Adult migrant children's perceived responsibility for providing elderly support.

| Category | Elderly | | Children | | Government | |
|---|---|---|---|---|---|---|
| | Mean | SD | Mean | SD | Mean | SD |
| Together in urban areas | 2.60 | 0.51 | 2.40 | 0.55 | 2.20 | 0.43 |
| Separately in urban areas | 2.28 | 0.45 | 2.26 | 0.47 | 2.10 | 0.29 |
| Together in rural areas | 2.62 | 0.64 | 2.38 | 0.59 | 2.14 | 0.41 |
| Separately in rural areas | 2.25 | 0.49 | 2.35 | 0.52 | 2.20 | 0.39 |
| Status quo | 2.41 | 0.50 | 2.27 | 0.54 | 2.27 | 0.53 |
| Nursing home | 2.50 | 0.56 | 2.16 | 0.58 | 2.09 | 0.38 |
| Care by siblings or relatives | 2.42 | 0.54 | 2.23 | 0.50 | 2.17 | 0.41 |
| Total | 2.54 | 0.52 | 2.34 | 0.53 | 2.17 | 0.41 |
| F-value | 11.24 | | 3.40 | | 8.63 | |
| *p*-value | 0.000 | | 0.009 | | 0.000 | |

Note: dependent variable = intended living arrangement for the left-behind rural parents.

All the mean values of attitude are significantly different among the seven groups, indicating that personal attitude toward "who is responsible for providing elderly support" is a potentially important factor influencing migrants' future behaviours regarding supporting their elderly parents. Table 5 demonstrates that those who chose the "co-residence" pattern scored relatively higher on "elderly" and "children" than the other subgroups, meaning that they considered that both "elderly" and "children" should be primarily responsible for old-age support. It is a traditional attitude rooted in Chinese society that rural populations themselves and family members guarantee the basic care of the elderly. The result might also reflect mistrust in governments.

As shown in Table 5, those who intended to put their parents in a nursing home were significantly more inclined to indicate that "elderly" rather than "children" are primarily responsible for elderly care. That is probably because their low financial capacity restrained the spirit of filial piety. Nevertheless, it is surprising that some members of this subgroup were high-paid and in high-end occupations. According to the traditional notion, putting parents in a nursing home is not a dutiful behaviour. As one respondent expressed "putting parents in a nursing home betrays the concept of filial piety, which fellow villagers would criticize". Some rural migrants opt for such a living arrangement

for their parents, partly due to the influence of modernization. As Cheung and Kwan [36] indicated, modernization may have adverse effects on traditional social norms, such as "filial piety", in Chinese culture. In modern society, traditional values might be weakened by the spread of individualism.

Compared with members of the other subgroups, those who planned to maintain the status quo emphasized that governments should take much more responsibility for elderly care. According to the interviews, those respondents tended to maintain their current pattern of living in cities for a variety of reasons. Some mentioned that the pressure of living in cities was so high that they could not support their elder parents. In contrast, others said the parents' earnings (rural social insurance and income from agricultural work) can meet their basic requirement in their rural hometowns; hence, there is no need to change the current living arrangement and bear more responsibility for elderly care.

In summary, the intended living arrangement for the left-behind parents is the function of the objective factors, such as the migrants' socioeconomic status, current support pattern, and future plan for residency arrangement, as well as the subjective factors, such as the migrants' perceived responsibility for providing elderly support. These factors influence the migrants' intended living arrangements for their left-behind elderly via different pathways. It should be noted that these factors may be interrelated, and some of them may exert meditating effects. For instance, the migrants' socioeconomic status directly affects their plans for the elderly parents' residency arrangement, while it indirectly affects the plans by influencing their current support pattern and perceived responsibility for providing elderly support.

## 5. Conclusions

In the urbanization process, the number of left-behind rural elderly has increased rapidly in China. Researchers and policymakers have expressed concerns about this social group due to rural China's incomplete social safety nets. Most academic studies address the living and caregiving arrangements for the rural elderly from the perspective of the left-behind elderly or the current support behaviours of migrant children. At the same time, little attempt has been made to address the intended living arrangements from the perspective of adult migrant children. To fill the above research gaps, this study was grounded on the 2015 questionnaire survey in Jiangsu province to investigate adult migrant children's intended living arrangement for their left-behind aging parents. The findings show that most adult migrant children (94.3%) intended to change the current left-behind status of their parents, while only 5.7% and 3.7% of the respondents wanted to maintain the status quo or send their parents to nursing homes, respectively. The substantial changes in the living arrangements between future plan and current situation confirm this study's necessity, which also profoundly impacts the healthcare and social insurance systems in rural areas.

It was found that adult migrant children's future expectations, care attitudes towards their parents, cultural embedding and living costs are closely associated with their intended living arrangements for their aging parents. First, distinct socioeconomic characteristics related to future expectations affect the rural migrant children's intended living arrangements for their left-behind parents. Briefly, younger rural migrants with more human capital prefer the modern living pattern and would like to bring their aging parents to urban areas because this social group usually has higher future expectations of settling down in cities and supporting their aging parents in an urban setting. In contrast, older rural migrants with less social capital would like to return to their rural hometowns to take care of their parents or to adopt negative support patterns such as simply allowing their parents to remain in rural areas. Moreover, the income and occupation levels of rural migrants are also important factors that influence their intentions of living arrangements. Normally, rural migrants with higher earnings and occupation ranks are more likely to pursue their desired support methods rather than passively choose other living patterns.

In brief, rural migrants with advantaged or disadvantaged conditions in cities would have distinct living arrangements with elder parents related to old-age support.

Second, the adult migrant children's intended living arrangements for their parents could be partly attributable to their attitudes toward elderly care. Most rural migrants still hold to traditional beliefs, such as providing family-based care to the elderly, co-residing with their parents, sending remittance to their parents, and keeping the ties with their parents through the use of new communication technologies. Nevertheless, the rapidly changing socioeconomic environment could profoundly impact migrant children's conception of elderly care. For instance, due to the increasing influence of individualism in modern culture, a few adult migrant children expressed their thought of not providing elderly care. They emphasized the provision of elderly support as the responsibility of the elderly themselves and the government rather than the children. Consequently, some adult migrant children neither share their earnings with nor provide emotional support to their left-behind parents, even with new communication technologies available.

Third, the Confucian doctrine is still pervasive among migrant children. A vast majority of adult migrant children intended to change the current left-behind status of their parents to fulfil their moral obligations. Most of them chose to care for their elderly parents in traditional ways, such as co-residing, either with themselves or their siblings. Moreover, married females intended to arrange for their parents to reside with their brothers, which is a manifestation of a patriarchal culture (i.e., the sons should be responsible for supporting the aging parents). The results indicate that although China is undergoing modernization, the country will probably only follow the Western pattern partially. Co-residence is still the dominant option for rural elderly support in China. Modernization is probabilistic, not deterministic [37], meaning that even though modernization is going to transform society, the process and the path have not defaulted; instead, they are contingent upon the historical and cultural context of the society concerned. The sociocultural context can have enduring influences on society, even under modernization. This topic is worth investigating in the future.

Fourth, living costs in urban areas are vital in shaping migrant children's intended living arrangements for their parents. In past decades, uneven development in China has driven the young working population to move from rural to urban areas and from the western to the eastern regions. Despite moving to more developed areas, rural migrants are often marginalized. During the investigation, it was observed that rural migrants were restricted by rigid institutional barriers and limited human capital compared to their urban peers. In addition to the institutional barriers, their residency choices were further confined by the unreasonably high living costs, especially the skyrocketing price of property and rental expenses in urban areas. In the rapidly changing urban areas, rural migrants try to fully utilize their limited socioeconomic capital to implement their family support strategies. The soaring living costs in large cities in the eastern region largely exhausts the financial capital of rural migrants to take care of their parents. To pursue better quality of life for their families and themselves, some of them chose to return to cities or towns near their rural hometowns, where they are more likely to have more preferable living arrangements for their parents (i.e., living separately with parents in the same city) and closer family ties. Those eager to settle in cities but with less human capital tended to choose the practical co-residence pattern or maintain the status quo. To make the most of the chances available to them, these migrant children planned to co-reside with their parents to save living costs in cities or chose not to support their aging parents. Overall, those living arrangement intentions associated with the residency plans of adult migrant children could reshape the geographic distribution of rural migrants and create a new trend of urban growth in China in the coming decades.

Although family-based attitudes and behaviours toward elderly care are still pervasive among rural migrants in contemporary China, institutional reforms related to elderly support should be implemented quickly. On the one hand, the low-level agricultural earnings and the unclear property rights of rural landholdings cannot fully guarantee the

basic living needs of the rural elderly. Meanwhile, the old-age pension and healthcare systems in rural China are still insufficient and cannot provide the rural elderly with adequate care [27]. Hence, the rural elderly have to rely on traditional forms of caregiving. They have a higher risk of suffering poverty and poor health if they receive little support from their children. On the other hand, the current welfare system continues to place the financial burden of elder care on the shoulders of adult children. Although the traditional thought of supporting aging parents is still dominant, rural migrants are often marginalized in cities mainly due to institutional barriers, high living costs and insufficient socioeconomic capital. Additionally, such thought has been challenged by modernization in many ways. This may impede proper living arrangements and old-age care provision for the rural elderly. Under such circumstances, it is urgently necessary to improve the current social welfare system and introduce more comprehensive institutional reforms.

One of the data limitations is that the locations of the migrants' origins was not recorded, which is believed to have a significant influence on their intended living arrangement intentions for their parents. Therefore, this study only examines the elderly parents' living arrangements from the migrants' perspective. Indeed, some of the left-behind parents may be quite active and capable of deciding their living arrangements, but this is not taken into account in this study. The above issues should be further addressed in future research.

**Author Contributions:** Writing—review and editing, validation and supervision, J.F.; Conceptualization, methodology, formal analysis, data curation and writing—original draft preparation, S.T. All authors have read and agreed to the published version of the manuscript.

**Funding:** This research was funded by the National Natural Science Foundation of China (41871135, 42071179).

**Institutional Review Board Statement:** Not applicable.

**Informed Consent Statement:** Not applicable.

**Data Availability Statement:** The datasets used and/or analyzed during the current study are available from the corresponding author upon reasonable request.

**Conflicts of Interest:** The authors declare no conflict of interest.

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
