# Peer review of "Living Arrangement Intentions of Adult Migrant Children toward Their Left-Behind Rural Parents in China"

_land, doi:10.3390/land12030526_

Round 1
Reviewer 1 Report
Dear Author, comments in the attachment.

Reviewer 2 Report
The paper used the qualitative analysis method to study urban immigrants' intentions for arranging for their left-behind parents. Questionnaire surveys were carried out in Suzhou and Nanjing in China. The topic is of significant novelty. The study fills in literature gaps on one of the most critical issues in developing countries. The paper is structured well, and the results are presented clearly. However, the following issues need to be clarified before publication.
the literature review and methodology framework failed to discuss how the one-child policy and gender differences impact immigrants' decisions;
In terms of sampling strategy, I fail to see how spatial patterns influence the choice of study target; are there any differences between a construction worker in one district and another?
the authors should specify whether the study target is temporary immigrants or permanent urban residents, as this will significantly change the results. A person with Hukou or property in two cities may have different perceptions about the living situation of their parents; was the immigrant status of their parents taken into consideration when validating the questionnaire?
The paper explores the role of siblings but fails to clarify how siblings play a role in parents' support. For example, is there any difference between a person with or without siblings?
I failed to understand the "rural hometown" , "same village" and "urban area". I suspect urban area indicates city, rural hometown is about county seat or township?
Moreover, table 2 appears to be an error. Not sure what the table is about
I recommend adding a discussion section and keeping the conclusion brief.
Round 2
Reviewer 2 Report
The authors have addressed the concerns from previous round review. The paper was revised. It is recommended publish after final language and reference check.
Author Response
Thank the reviewer. We have carefully checked the language and reference. Please see the changes in the manusript.